# Chemical modifications of adenine base editor mRNA and guide RNA expand its application scope

Tingting Jiang[1], Jordana M. Henderson[2], Kevin Coote[3], Yi Cheng[3], Hillary C. Valley[3], Xiao-Ou Zhang [4], Qin Wang [5], Luke H. Rhym[6,7], Yueying Cao[1], Gregory A. Newby[8,9,10], Hermann Bihler [3], Martin Mense [3], Zhiping Weng[4], Daniel G. Anderson [6,7,11,12], Anton P. McCaffrey[2], David R. Liu[8,9,10] & Wen Xue [1,13,14,15 ✉]

CRISPR-Cas9-associated base editing is a promising tool to correct pathogenic single nucleotide mutations in research or therapeutic settings. Efficient base editing requires cellular exposure to levels of base editors that can be difficult to attain in hard-to-transfect cells or in vivo. Here we engineer a chemically modified mRNA-encoded adenine base editor that mediates robust editing at various cellular genomic sites together with moderately modified guide RNA, and show its therapeutic potential in correcting pathogenic single nucleotide mutations in cell and animal models of diseases. The optimized chemical modifications of adenine base editor mRNA and guide RNA expand the applicability of CRISPR-associated gene editing tools in vitro and in vivo.

[1] RNA Therapeutics Institute, University of Massachusetts Medical School, Worcester, MA 01605, USA. [2] TriLink BioTechnologies, San Diego, CA, USA. [3] Cystic Fibrosis Foundation, CFFT Lab, Lexington, MA 02421, USA. [4] Program in Bioinformatics and Integrative Biology, University of Massachusetts Medical School, Worcester, MA, USA. [5] School of Life Sciences and Technology, Tongji University, 200092 Shanghai, China. [6] David H. Koch Institute for Integrative Cancer Research, Massachusetts Institute of Technology, Cambridge, MA, USA. [7] Department of Chemical Engineering, Massachusetts Institute of Technology, Cambridge, MA, USA. [8] Merkin Institute of Transformative Technologies in Healthcare, Broad Institute of Harvard and MIT, Cambridge, MA, USA. [9] Howard Hughes Medical Institute, Harvard University, Cambridge, MA 02138, USA. [10] Department of Chemistry and Chemical Biology, Harvard University, Cambridge, MA 02138, USA. [11] Institute for Medical Engineering and Science, Massachusetts Institute of Technology, Cambridge, MA, USA. [12] Harvard-MIT Division of Health Sciences & Technology, Massachusetts Institute of Technology, Cambridge, MA, USA. [13] Department of Molecular, Cell and Cancer Biology, University of Massachusetts Medical School, Worcester, MA, USA. [14] Department of Molecular Medicine, University of Massachusetts Medical School, Worcester, MA, USA. [15] Li Weibo Institute for Rare Diseases Research, University of Massachusetts Medical School, 368 Plantation Street, Worcester, MA 01605, USA. ✉email: Wen.Xue@umassmed.edu

G·C-to-A·T base pair substitution accounts for approximately half of known pathogenic single nucleotide mutations in humans[1]. Adenine base editor (ABE), which is constructed by fusing adenine deaminase to catalytically inactive CRISPR-associated (Cas) protein, can precisely and permanently convert A·T to G·C with the guidance of target specific guide RNA without creating a double-strand DNA break or requiring an exogenous DNA repair donor[1,2]. Thus, ABE may be a useful tool to model or treat genetic diseases that are associated with single nucleotide mutations. Indeed, many ABE variants have been widely studied and applied[3–9]. However, high cellular expression of ABE agents is required for effective base editing[10,11], which limits the application of ABE plasmids in cells that are difficult to transfect.

To date, the therapeutic potential of ABE has been demonstrated by delivery of DNA-encoded base editors to adult animal disease models via plasmids[4] or AAVs[9,12,13]. However, these approaches raise the potential for DNA integration or off-target effects due to long-term exposure to the gene editing machinery, hindering their clinical relevance. Developing non-DNA-encoded base editor and non-viral delivery methods may facilitate broader application of ABE.

Microinjection of cytidine base editor mRNA and in vitro transcribed guide RNA can introduce effective base editing in mouse embryo or pig oocyte[14,15], suggesting the potential of using RNA-encoded base editors to mediate editing. However, the editing efficiency mediated by RNA-encoded ABE system has not been studied in somatic cells, and its application potential and delivery method have not been addressed.

Here, we engineer an RNA-encoded ABE system by introducing various chemical modifications to both ABE mRNA and guide RNA. The optimized base editing system exhibits higher editing efficiency at some genomic sites compared to DNA-encoded system. Furthermore, we demonstrate that this RNA-based system mediates robust editing in hard-to-transfect cystic fibrosis bronchial epithelial cells. Moreover, by encapsulating modified ABE mRNA and guide RNA into lipid nanoparticle (LNP), we successfully deliver the RNA-encoded ABE into the liver of Tyrosinemia I mice, correct the disease mutation, and rescue the phenotype. Our engineered RNA-encoded system expands the application scope of base editors.

## Results

### Uridine depleted ABE mRNA modified with 5-methoxyuridine.
We recently engineered a codon-optimized variant of ABE ("RA6.3"), recognizing "NGG" PAM sequence, with improved editing efficiency in HEK293T cells[4]. Hydrodynamically-injected plasmids delivered RA6.3 and guide RNA to mouse liver and corrected a splice-site mutation of fumarylacetoacetate hydrolase (*Fah*) gene. Yet, when we tested a non-viral delivery method of RA6.3 in vivo, we found that lipid nanoparticle (LNP)-mediated delivery of unmodified mRNA supported three-fold lower editing efficiency compared to plasmid-delivered RA6.3 in Tyrosinemia I mice[4]. This lower editing rate may be due to instability of unmodified RA6.3 mRNA, and subsequently, poor expression in cells. Indeed, protein expression level by transiently transfecting unmodified RA6.3 mRNA is much lower than that from the well-characterized and widely-applied chemically modified Cas9 mRNA[16,17] (Supplementary Fig. 1a). To increase its cellular expression, we set out to optimize the chemical composition of ABE mRNA.

Based on the primary structure of RA6.3 (Supplementary Fig. 1b) as well as the reports that 5′ capping stabilizes mRNA[18] and uridine substitution or modification increases Cas9 mRNA activity[16], we engineered three versions of 5′ capped RA6.3

mRNA: unmodified, uridine-depleted, and 5-methoxyuridine-modified (5moU) (Fig. 1a). For uridine-depleted mRNA, we used synonymous codons to deplete the transcript of as many uridines as possible without altering the coding sequence. 5moU mRNA was derived from uridine-depleted mRNA, and replaced all remaining uridines with 5-methoxyuridine. After transient transfection into HEK293T cells, only 5moU mRNA (hereafter 5moU-6.3) yielded stable protein expression comparable to Cas9 mRNA (Fig. 1b). The expression dynamics of 5moU-6.3 were also similar to Cas9 mRNA—detectable at 6 h post-transfection, reached peak expression after 1 day, and degraded after 2 days (Supplementary Fig. 1c). Thus, we elected to use our modified RNA-encoded ABE, 5moU-6.3, for subsequent experiments to test the editing efficiency of an RNA-encoded ABE system.

### Moderately modified gRNA mediates robust editing.
The guide RNA of CRISPR-Cas9-associated editing system consists of a CRISPR RNA (crRNA) and a transactivating crRNA (tracrRNA). After electroporating 5moU-6.3 and unmodified tracrRNA:crRNA to HEK293T cells, we observed minimal editing (<1%; Supplementary Fig. 1d, e) at a genomic site where DNA-encoded RA6.3 and guide RNA mediated an average A-to-G conversion rate of 19.5% as described before[4]. Previous reports suggest that chemical modifications improve guide RNA stability[19–21]. To improve the editing efficiency of our RNA-encoded ABE system, we designed moderately modified or heavily modified[20] tracrRNA and crRNA (Supplementary Fig. 1d), and co-electroporated them with 5moU-6.3 to compare editing efficiencies. Moderately modified tracrRNA:crRNA (2′-O-methyl 3′-phosphorothioate modification at first and last three nucleotides) conferred the highest editing efficiency (Supplementary Fig. 1e, f), and was comparable to guide RNA expressed from plasmid (Fig. 1c). Because plasmid-expressed guide RNA is single guide RNA, we then designed moderately modified single guide RNA (sgRNA) (Supplementary Fig. 1g). 5moU-6.3 and modified sgRNA could mediate the A-to-G conversion rates comparable to RA6.3 and sgRNA expressed from plasmids[4] (Fig. 1c). We also tested the editing efficiency of 5moU-6.3 and moderately modified sgRNA at two genomic sites where DNA-encoded RA6.3 mediates limited conversion rates. Our RNA-encoded ABE system showed significantly higher editing rates at all "A" sites within the editing window (Fig. 1d, e). Next, we compared the editing efficiency between unmodified and 5-methoxyuridine-modified ABE mRNA, when delivered with moderately modified sgRNA. We first measured A-to-G conversion rate at the same genomic site as in Fig. 1c at different concentrations (ranging from 0.0015 to 0.5 μg) of unmodified-6.3 or 5moU-6.3 (Supplementary Fig. 2a). We found that, at lower dosages (less than 0.015 μg), 5moU-6.3 shows ~1.5-fold higher editing efficiency compared to unmodified mRNA. Similarly, at two other genomic sites (same sites as in Fig. 1d, e), 5moU-6.3 mediated substantially higher A-to-G conversion rates than unmodified mRNA (Supplementary Fig. 2b). These data demonstrate a robust chemically modified RNA-encoded system for base editors.

To compare the off-target effects raised by DNA versus modified RNA-encoded ABE systems, we analyzed A-to-G conversion rates at the top known off-target loci for the two guide RNAs[22,23] used in Fig. 1d, e. Deep sequencing data show that, overall, base editing rates at the off-target sites are low (<0.2% in all groups) (Supplementary Fig. 3a, b). A significant increase in A-to-G conversion rate was only detected at 2 of the 14 "A" sites in 5moU-6.3-treated cells compared to control or DNA-encoded ABE-treated cells. Our results suggest that chemically-modified ABE mRNA can improve on-target editing

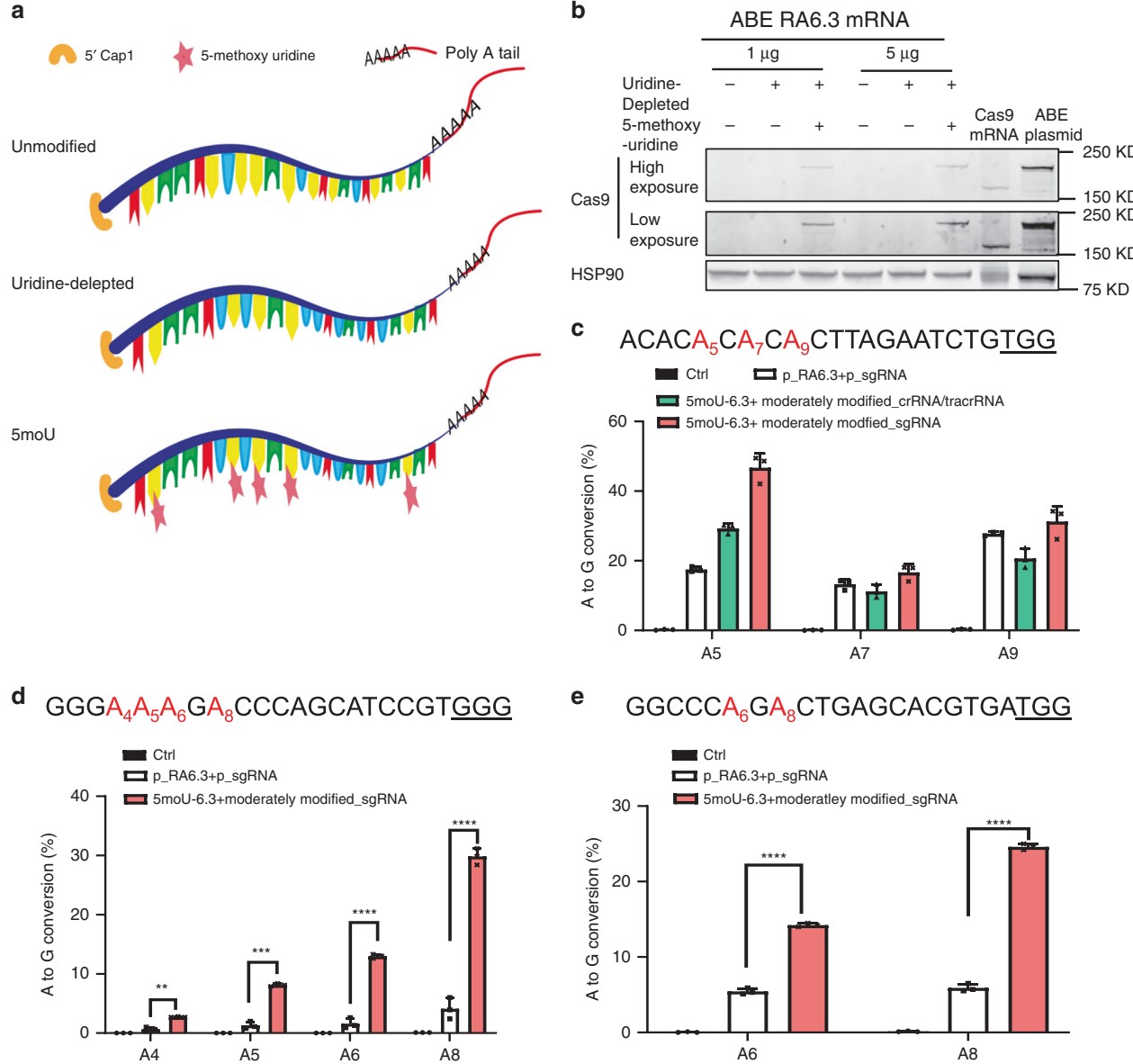

**Fig. 1 Chemical modifications of ABE mRNA and guide RNAs are critical for efficient base editing in cells. a** Diagrams of ABE RA6.3 mRNAs with sequence-optimization (uridine-depletion) and chemical modification (5-methoxyuridine). Red: "A"; Yellow: "U"; Green: "G"; Blue: "C". **b** ABE mRNA with both 5-methoxyuridine modification and uridine-depletion showed the highest protein expression in HEK293T cells by western blot. The mass of transfected RA6.3 mRNA is as indicated. ABE plasmid and Cas9 mRNA served as controls. **c** Comparison of editing efficiency by plasmids of ABE and guide RNA (p_RA6.3 + p_sgRNA), ABE mRNA + tracrRNA/crRNA, and ABE mRNA + sgRNA in HEK293T cells. The target "A" sites are highlighted in red. PAM is underlined. Data represent mean ± SD. **d**, **e**, Comparison of editing efficiency by DNA and RNA-encoded ABE at two genomic sites in HEK293T cells. The target "A" sites are highlighted in red. PAM is underlined. **c–e** All mRNAs are 5moU-6.3. All guide RNAs are moderately modified RNA. Control group (Ctrl) is cells transfected with 500 ng GFP plasmid. Graphs show mean values. Data represent mean ± SD ($n = 3$ biologically independent samples). **d** $P = 0.0079$ (**), $0.0005$(***), $<0.0001$ (****), $<0.0001$ (****); **e** $P < 0.0001$ (****) (two tailed $t$-test). Source data are provided as a Source Data file for **b–e**.

compared to DNA-encoded ABE without substantially increasing off-target effects.

**ABE corrects a nonsense mutation in cystic fibrosis model.** Next, we investigated whether our engineered RNA-encoded ABE system can correct a pathogenic single nucleotide mutation in a cystic fibrosis (CF) cell model. Approximately 10% of CF patients carry cystic fibrosis transmembrane conductance regulator (CFTR) nonsense mutations that cannot be treated with any FDA-approved CFTR modulators, and around half of these mutations are correctable by ABE[24] (Supplementary Fig. 4a). The

second most common CFTR nonsense mutations, W1282X[25,26], is caused by a G>A mutation in exon 23 that produces minimal amount of functional CFTR protein and abolishes its Cl− transport activity[27]. A previous report showed that the human bronchial epithelial cell line (CFF-16HBEge CFTR W1282X), homozygous for W1282X mutation, recapitulates the phenotype of primary W1282X CF cells[25], making it a good cell model to study the correction of CFTR mutation. SV40 (polyomavirus simian virus 40), which was used to immortalize the 16HBEge cell line's parental 16HBE14o- cells, is integrated into one of the CFTR alleles[28]. The CFTR allele with SV40 incorporation does

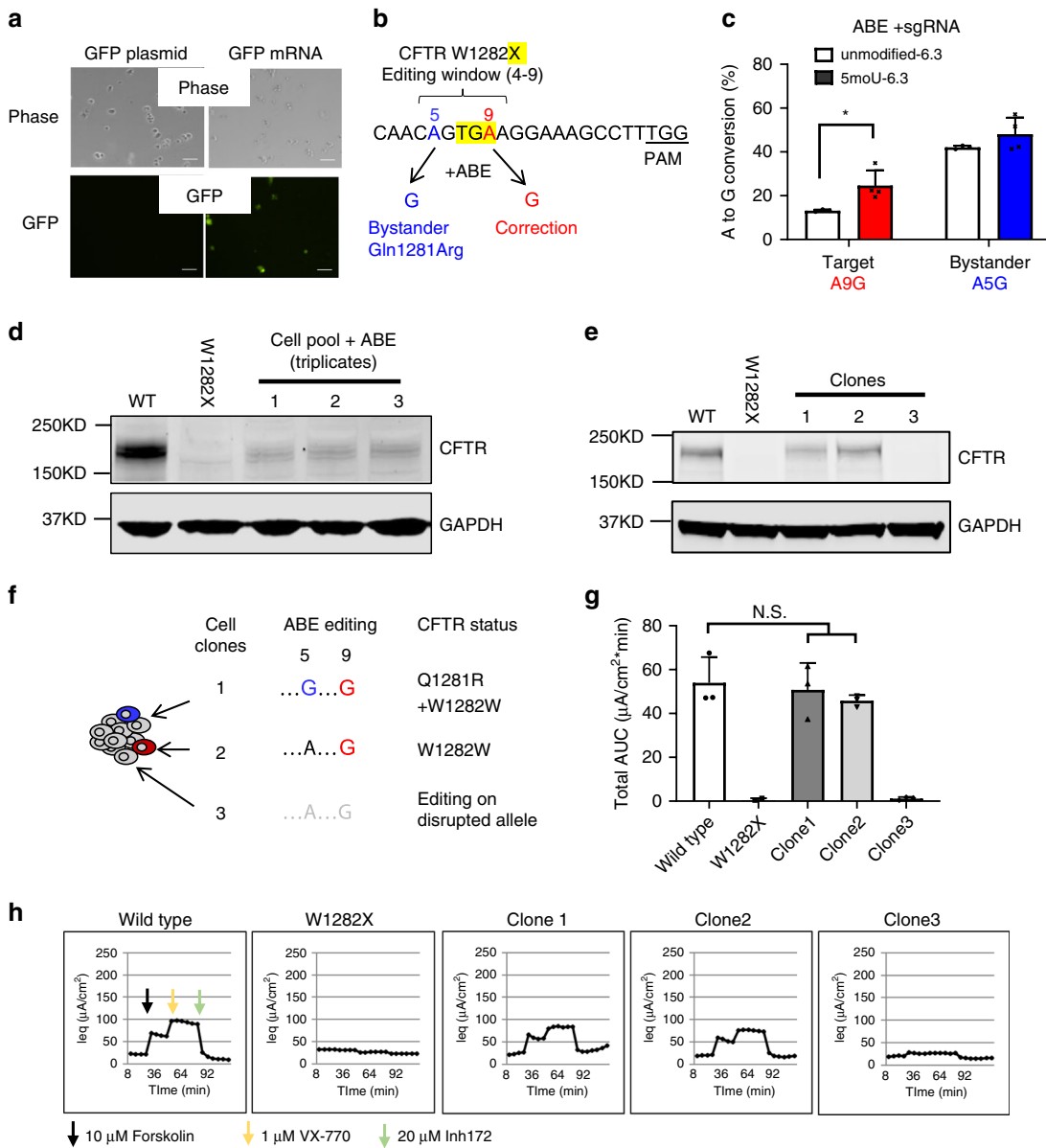

**Fig. 2 RNA-encoded ABE restores CFTR function in human lung airway cells with W1282X mutation. a** Expression of GFP plasmid and mRNA in CFF-16HBEge W1282X cells. Twelve hours of post electroporation, bright field and fluorescence images were taken. Scale bar = 100 μm. **b** Protospacer sequence used to base edit W1282X. The pre-mature stop codon is highlighted in yellow. PAM sequence is underlined. Target "A9" is red and bystander "A5" is blue. **c** A-to-G conversion rate in W1282X cell pools electroporated with moderately modified sgRNA and either unmodified (unmodified-6.3) or modified (5moU-6.3) ABE mRNA. Data represent mean ± SD ($n = 3$ (unmodified-6.3), $n = 4$ (5moU-6.3) biologically independent experiments). *$P = 0.0456$. **d** CFTR protein expression in 5moU-6.3-treated W1282X cell pools by western blot. GAPDH is loading control. Experiments were done for three times, and one is shown. **e** CFTR protein expression in three single cell clones. **f** Genotypes of cell clones. Clone 1 has a bystander editing. Clone 3 has base editing only on the SV40 disrupted CFTR allele. **g** ABE restored CFTR-mediated Cl- transport in clones 1 and 2. Values of area under the curve (AUC) in h is shown. Data represent mean ± SD ($n = 3$ wells of cells examined in one experiment). N.S. not significant (two tailed $t$-test). **h** Representative traces from electrophysiology assays. Source data are provided as a Source Data file for **c–e**.

not produce protein[29]. Unlike other cells (Supplementary Fig. 4b), 16HBE14ge cells poorly express genes that are delivered by plasmid (Fig. 2a). In contrast, electroporated mRNA expressed well in this cell line, suggesting that we could test our RNA-encoded ABE system to correct CFTR W1282X mutation. RA6.3 could correct this mutation by recognizing the "TGG" PAM sequence, and the target "A" site falls at position 9 of a protospacer that RA6.3 could use to correct this mutation, just at the upper edge of the editing window (Fig. 2b). We electroporated modified sgRNA with either unmodified-6.3 or 5moU-6.3 into 16HBEge cells to compare editing efficiency of unmodified versus

5-methoxyuridine-modified ABE mRNA. 5moU-6.3 achieved a significantly higher A-to-G conversion rate ($26.4 ± 7.40$%) at A9 target site compared to unmodified-6.3 ($13.1 ± 0.509$%). Furthermore, 5moU-6.3 restored full length CFTR protein expression to ~10% of the level in wild-type cells (Fig. 2d).

Notably, the editing efficiency at the bystander "A" site at position 5 (Fig. 2b; $45.1 ± 5.66$%) was higher than the target A9 site (Fig. 2c), and was associated with the corrected CFTR allele (Supplementary Fig. 4c). This bystander mutation changes codon 1281 from glutamine (Q) to arginine (R). To investigate whether this amino acid alteration affects CFTR function, we

isolated three single-cell clones with or without the bystander mutation. Each clone had one ABE-corrected *CFTR* allele (~50% A-to-G conversion at A9; Supplementary Fig. 4d). Clone 1 contained the bystander mutation on the corrected allele, clone 2 contained the bystander mutation on the uncorrected allele, and clone 3 did not contain bystander editing at majority (~80%) of corrected alleles (Supplementary Fig. 4e). Because SV40 disrupts the protein expression of one *CFTR* allele[29], we measured CFTR protein expression in each of these edited clones. Clones 1 and 2 had fully restored CFTR expression compared to wild-type cells, but clone 3 showed no CFTR expression (Fig. 2e). We conclude that the corrected target site in clone 3 is on the SV40-disrupted allele (Fig. 2f and Supplementary Fig. 4f). Next, we evaluated Cl⁻ channel activity mediated by ABE-edited CFTR and found that the cells in clones 1 and 2 exhibited similar Cl⁻ channel activity comparable to parental 16HBE41o- cells expressing wild type CFTR (Fig. 2g, h). This suggests that ABE-corrected CFTR has normal function and the bystander Q1281R mutation does not affect CFTR function. The cells in clone3 did not have any restored CFTR function. Our findings demonstrate that RA6.3 can correct a CF mutation, and RNA-encoded system could mediate robust adenine base editing in hard-to-transfect cells.

**ABE corrects a splice site mutation in Tyrosinemia I mouse model.** Finally, we tested the non-viral delivery of adenine base editor to the liver of a mouse model of Tyrosinemia I. Tyrosinemia I mice harbor a homozygous G•C to A•T point mutation in the last nucleotide of exon 8 in the *Fah* gene, causing exon 8 skipping, FAH protein deficiency, and liver damage. To maintain body weight and survival, these mice are given water supplemented with NTBC [2-(2-nitro-4-trifluoromethylbenzoyl)-1,3-cyclohexanedione], a tyrosine catabolic pathway inhibitor. RA6.3 can correct the causative mutation using a protospacer that places the target nucleotide at position 9 (Fig. 3a). Hepatocytes with corrected FAH protein will gain growth advantage and eventually repopulate the liver[30]. We used LNP to separately encapsulate 5moU-6.3 and modified sgRNA. After confirming LNP-delivered 5moU-6.3 could be expressed in cells (Supplementary Fig. 5a), we injected LNP via tail vein into adult mouse liver for four dosages based on reported studies[4,20]. One week after the last injection, we replaced NTBC-supplemented water with normal water to allow the repopulation of corrected cells. Treated mice maintained their body weight, while the untreated mouse rapidly lost its body weight (Fig. 3b), suggesting restoration of FAH function in LNP-treated mice. We observed widespread FAH-positive patches in mouse liver by staining with a FAH-specific antibody (Fig. 3c), and protein restoration rate was comparable to that in mice treated with plasmid-delivered RA6.3 via hydrodynamic injection (Supplementary Fig. 5b). Next, we investigated efficiency of *Fah* gene correction. On the transcriptome level, RT-PCR results from liver tissues of LNP-treated mice revealed that the majority of *Fah* mRNA was now properly spliced (Fig. 3d). On the DNA level, we observed an A-to-G correction rate of $12.5 \pm 2.67\%$ at the target site in the liver tissues of LNP-treated mice (Fig. 3e), which is comparable to our previously reported plasmid-delivered ABE system ($9.5 \pm 4.0\%$)[4]. Notably, there is a bystander site within the editing window (A6, Fig. 3a). The A-to-G conversion at this site changes a serine codon into alanine (S235A) in the FAH enzyme. Because S235 is near the FAH enzyme active site[31], A-to-G conversion at A6 site will not rescue the splicing defect but may affect enzyme activity[4]. Surprisingly, at this bystander site, we observed a significantly lower editing rate ($0.096 \pm 0.032\%$) in LNP-treated mice compared to our previously reported rate ($1.9 \pm 0.9\%$)[4] ($P = 0.0006$, Supplementary Fig. 5c), suggesting that the short half-life of ABE

mRNA might minimize bystander conversions in vivo. This LNP-mediated non-viral delivery of modified RNA-encoded ABE, unlike hydrodynamic injection of plasmid DNA, provides a clinically-relevant therapeutic method for genetic diseases caused by single nucleotide mutations.

## Discussion
By applying various chemical modifications to ABE mRNA and guide RNA, we successfully engineered an RNA-encoded base editing system. Although previous reports show that unmodified cytidine base editor mRNA and guide RNA could effectively edit embryos and oocytes[14,15], our data show that unmodified ABE mRNA does not effectively express in HEK293T cells and unmodified guide RNA cannot mediate efficient editing in somatic cell culture. This might be because unmodified mRNA and guide RNA are not stable and will quickly undergo degradation after being delivered to cell culture. We further demonstrate that chemical modifications are essential for RNA-encoded base editor to mediate efficient editing. It has been reported that uridine depletion and pseudo-uridine could increase the activity of Cas9 mRNA and reduce immune responses elicited by Cas9 mRNA[16]. In this study, we demonstrate that uridine depletion and 5-methoxyuridine modification is critical for the stable expression of ABE mRNA. Furthermore, we expect this optimized RNA system will be applicable for other CRISPR-associated editors, e.g., primer editors[22].

We did not observe significant off-target effects raised by our modified mRNA-encoded ABE at top known off-target sites for guide RNAs. Some off-target events of base editors, on both the DNA and RNA level, may be independent of guide RNA sequence[5,14,32–34]. Therefore, future work should perform unbiased screens to detect off-target editing by modified ABE mRNA at the whole genome and transcriptome levels.

Unlike in the well-established CRISPR-Cas9 field, to date studies of using protein or RNA-encoded base editor to correct diseases have been exceedingly limited[35]. Here, we provide the first report on delivery of RNA-encoded ABE to effectively correct disease-causing point mutations in vitro and in vivo. The successful correction of an "untreatable" CF mutation demonstrates the potential of base editing as a gene therapy method for CF treatment. However, due to that the incorporation of SV40 disrupts the expression of one of the *CFTR* alleles[28], the effect of 5moU-6.3 on correcting CF phenotype (protein expression and electrophysiological function) is likely underestimated in our study. A recent study has used base editor to correct CFTR nonsense mutations in organoid of cystic fibrosis patients[36]. Future work should apply base editing to correct primary CF cells or animal models.

LNP-based non-viral delivery of ABE successfully correct a *Fah* point mutation in Tyrosinemia I mice, providing a clinically-relevant method to treat genetic diseases with base editing. Corrected cells with normal FAH function have a growth advantage and outgrow the non-corrected cells, magnifying the therapeutic effect. To provide a delivery method suitable for treating a broad array of diseases, future work should optimize delivery dosage and nanoparticle formulations for base editor encapsulation to maximize initial editing efficiency.

In summary, our optimized ABE mRNA and guide RNA reagents unlock new therapeutic possibilities by using adenine base editing.

## Methods
**Cell culture**. Human embryonic kidney (HEK293T) cells (ATCC) were maintained in Dulbecco's Modified Eagle's Medium (Corning) supplemented with 10% fetal bovine serum (Gibco) and 1% Penicillin/Streptomycin (Gibco). CFF-16HBEge CFTR W1282X cells were obtained from the Cystic Fibrosis Foundation's

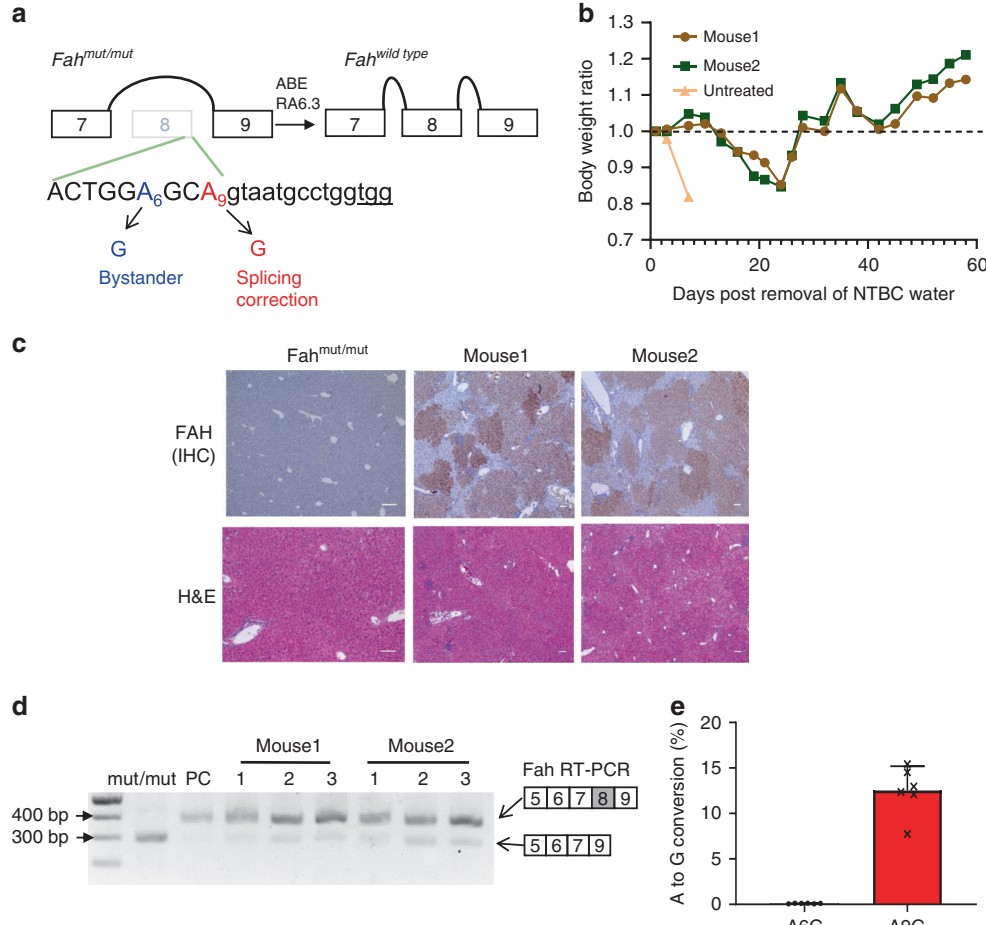

**Fig. 3 Nanoparticle delivery of ABE mRNA to correct Fah splice-site mutation in vivo. a** Diagram of Fah splicing before and after correction by RA6.3. Protospacer sequence is shown below. Exon sequence is in upper case and intron sequence is in lower case. Target "A" is red and bystander "A" is blue. PAM sequence is underlined. **b** Tracking mice body weight ratio after removal of NTBC water. Body weight ratio is calculated over the body weight at day 0 of NTBC removal. **c** Immunohistochemistry staining and Hematoxylin and Eosin staining (H&E) of mouse liver sections. Mouse 1 and 2 denotes mice treated with LNP-5moU-6.3 and LNP-sgFah (end point 58 days). Untreated Fah mut/mut mouse was kept on NTBC water. Scale bar = 100 µm **d** RT-PCR results from treated mouse liver. PC (positive control) indicates samples from mouse treated with plasmids-delivered ABE and guide RNA through hydrodynamic injection. Three liver lobes (as 1, 2, and 3) per mouse were collected and analyzed. Wildtype *Fah* amplicon is 405 bp and mutant *Fah* (lacking exon 8) is 305 bp. **e** A-to-G conversion rate at *Fah* gene locus in mice livers. Three liver lobes per mouse were collected and analyzed. A6G/A9G indicates editing efficiency at bystander/target "A" sites. Data represent mean± SD ($n = 6$). Source data are provided as a Source Data file for **b**, **d**, **e**.

Therapeutic Lab (Lexington, MA). Parental 16HBE14o-cells expressing wild type CFTR were from Millipore. Cells were cultured in Minimum Essential Medium (Corning) supplemented with 10% fetal bovine serum (Gibco) and 1% Penicillin/Streptomycin (Gibco). Flasks were pre-coated by incubating with a thin layer of coating solution (LHC-8 basal medium (Thermo Fisher), 1.34 µl/ml Bovine serum albumin 7.5% (Thermo Fisher), 10 µl/ml Bovine collagen solution (Thermo Fisher), Type 1, 10 µl/ml Fibronectin from human plasma (Advanced Biomatrix)) at 37 °C/5% $CO_2$ for 3 h. All the cells were incubated at 37 °C in a humidified 5% $CO_2$ atmosphere.

**Chemically modified mRNA and guide RNA.** Uridine depletion was carried out using the "optimize codons" tool in Geneious version R8.0.5 (https://www.geneious.com). A new sequence in Geneious was created; this sequence was selected, and under the tab "annotate and predict," the "optimize codons" function was chosen. Parameters were chosen as follows: source of genetic code, standard; target organism, Homo sapiens; target genetic code, standard; threshold to be rare = 1; and avoid restriction sites.

mRNAs were synthesized by T7 RNA polymerase in vitro transcription. Unmodified or 5-methoxyuridine modified mRNAs were co-transcriptionally capped to produce cap 1 mRNAs using the CleanCap® Reagent AG (TriLink BioTechnologies; Cat #: N-7113). All enzymes were purchased from New England Biolabs. Transcriptions were done in 1× transcription buffer (40 mM Tris, 10 mM dithiothreitol, 2 mM spermidine, 0.002% Triton X-100, and 16.5 mM magnesium acetate) using final concentrations of 8 U/µl T7 RNA polymerase (M0251L); 0.002 U/µl inorganic pyrophosphatase (M2403L); 1 U/µl murine RNase inhibitor (M0314L); 0.025 µg/µl standard or uridine-depleted transcription template; 4 mM

CleanCap Cap 1 AG trimer; and 5 mM each of adenosine triphosphate, cytidine triphosphate, guanosine triphosphate, and uridine triphosphate (or 5-methoxyuridine triphosphate; TriLink BioTechnologies; Cat #: N-1093), as indicated. Unmodified nucleoside triphosphates were purchased from Roche Diagnostics. Transcription reactions were incubated at 37 °C for 2 h and treated with final 0.4 U/µl DNase I (M0303L) in 1× DNase I buffer for 15 min at 37 °C. mRNAs were purified by RNeasy Maxi (QIAGEN, 75162), phosphatase treated for 1 h with final 0.25 U/µg Antarctic phosphatase (M0289L) in 1× Antarctic phosphatase buffer, and then re-purified by RNeasy. Transcription quality was measured by bioanalyzer analysis (Agilent 2100 Bioanalyzer).

Chemically modified Cas9 mRNA (5meC/pseudouridine)[17] was from TriLink. The unmodified RA6.3 mRNA (Supplementary Fig. 1a) was as described[4]. All the guide RNA sequences are listed in supplementary table. The modified single guide RNAs, moderately modified tracrRNA and crRNA were synthesized by Synthego. Unmodified tracrRNA and crRNA were synthesized by IDT. The heavily modified crRNA was designed according to a previous report[20] and synthesized by IDT.

**Transfection of HEK239T cells.** To validate the expression of RA6.3 mRNA in cells (Fig. 1b and Supplementary Fig. 1a, c), HEK293T cells were seeded at 70% confluence in 12-well cell culture plate one day before transfection. RA6.3 mRNA or plasmid was transfected using Lipofectamine 3000 reagent (Invitrogen). Specifically, in Fig. 1b, 1 µg Cas9mRNA, 0.5 µg RA6.3 plasmid or the indicated amounts of RA6.3 mRNA was transfected. In Supplementary Fig. 1a, 2 µg RA6.3 mRNA or 0.5 µg Cas9 mRNA was transfected. In Supplementary Fig. 1c, 1 µg 5moU-6.3 mRNA or Cas9 mRNA was transfected. In Fig. 1b, cells were lysed for western blot analysis 6 h after transfection of mRNA and two days after transfection of

plasmids. In Supplementary Fig. 1a, cells were lysed after an 8-h incubation following transfection.

**Electroporation of HEK293T cells and 16HBEge cells**. The Neon Transfection System (Invitrogen) was used for electroporation. To assess and compare editing efficiency of ABE delivered as DNA or RNA (Fig. 1c–e and Supplementary Fig. 1e, f), 2 μg RA6.3 plasmid and 1 μg guide RNA plasmid or 3 μg 5moU-6.3 and 0.2 pmol of guide RNA was electroporated into $2 \times 10^5$ HEK293T cells. For the control group (Ctrl), 500 ng of a GFP plasmids were electroporated into HEK293T cells before seeding in a 12-well cell culture plate. To compare A-to-G conversion rates mediated by modified and unmodified ABE mRNA in HEK293T cells, 0.015 μg mRNA and 0.1 pmol guide RNA were electroporated into $3 \times 10^4$ cells (Supplementary Fig. 2b). HEK293T cells were electroporated using 1150 V, 20 ms and 2 pulses and seeded in 12-well plate. Three microgram of 5moU-6.3 and 0.2 pmol guide RNA was electroporated into $2 \times 10^5$ 16HBEge cells using 1300 V, 10 ms and three pulses and then all seeded in pre-coated 12-well plate. Sixteen hours of post electroporation, the cells were replaced with fresh culture medium.

**Western blot**. Post-transfected cells were lysed with RIPA buffer (Boston bioproducts) supplemented with protease inhibitor (Roche) and phosphatase inhibitor (Thermo Fisher). Protein concentration was measured by BCA assay kit (Thermo Fisher). Equal amounts of proteins were loaded onto NuPAGE™ 4–12% Bis–Tris Protein Gels (Invitrogen) and run at 125 V for 90 min. After being transferred to nitrocellulose membrane, the blots were incubated with indicated antibodies, anti-Cas9 (A-9000-050, Epigentek Group, dilution: 1:1000), anti-GAPDH (MAB374, EMD Millipore, dilution: 1:5000) or anti-CFTR (UNC-596, University of North Carolina at Chapel Hill, dilution:1:1000), followed by incubation with distinct fluorophore-conjugated secondary antibodies. The images were captured using Odyssey system (Li-Cor Biosciences).

**Mouse experiments**. All animal study protocols were approved by the UMass IACUC (University of Massachusetts Medical School institutional animal care and use committee). Fah[mut/mut] mice[30] were kept on 10 mg/l NTBC water. Lipid nanoparticle formulation and treatment was as previously described[20]. 1 mg/kg LNP-5moU-6.3 mRNA and 0.5 mg/kg LNP-Fah sgRNA were injected in 7-week-old female Fah[mut/mut] mice via tail vein injection and the mice were injected for four doses (once every 3 days) and kept on NTBC water during the treating period. For the positive control group (plasmid injected), 30 μg RA6.3 plasmid and 60 μg sgFah expressing plasmid were injected to mouse for one dose[4]. One week after the last injection, NTBC supplemented water was replaced with normal water. At this point, mouse weight was measured every two days. As per our guidelines, when the mouse lost 20% of its body weight relative to the first day of measurement (the day when remove NTBC supplemented water), mouse will be re-treated with NTBC supplemented water until the body weight is back to original body weight. After the body weight remained stable for 10 days without the need to be treated with NTBC supplemented water, mice were euthanized according to guideline.

**Genomic DNA extraction**. To extract the genomic DNA from HEK293T cells, cells (5 days post transfection) were washed with PBS, pelleted, lysed with 50 μl Quick extraction buffer (Epicenter), and incubated in a thermocycler (65 °C 15 min and 98 °C 5 min). $2 \times 10^5$ 16HBE14o- cells (wild type, W1282X, corrected pools and three single-cell clones) were used for extracting genomic DNA using PureLink Genomic DNA Mini Kit (Thermo Fisher). The same Kit was also used to extract genomic DNA from mouse liver tissues (~ 10 mg each), three samples (from different liver lobes) per mouse.

**Immunohistochemistry**. Part of Livers were fixed with 4% formalin, embedded in paraffin, sectioned at 5 μm and stained with hematoxylin and eosin (H&E) for pathology. Liver sections were de-waxed, rehydrated and stained using standard immunohistochemistry protocols[37]. The following antibody was used: anti-FAH (ab83770, Abcam, dilution:1:400). The images were captured using Leica DMi8 microscopy.

**RNA extraction and gene expression analysis using RT-PCR**. Liver tissues (~10 mg each) were used for extracting RNA (three samples from different lobes/ mouse). RNA was purified using Trizol (Invitrogen) and reverse-transcribed using High-Capacity cDNA Reverse Transcription Kit (Applied Biosystems). The amplicon is from Exon5 to Exon9 in the Fah transcript. Primers used in RT-PCR are listed in Supplementary Table.

**Identification of edited clones**. To grow single 16HBE14o- cell clones, the post-electroporated cells were serially diluted into 10 cells/ml in culture medium and seeded to pre-coated 96-well plates (100 μl medium/well). The plates were incubated at 37 °C in a humidified 5% CO₂ atmosphere. The medium was changed every three days. After 20 days, the cell colonies were dissociated with 50 μl TrypLE Express (Gibco) and re-seeded into matching wells of two 24-well pre-coated plates (Plate 1 and 2).

After cell confluence reached 30% in 24-well plate, cells in plate 1 were used for extracting genomic DNA with 30 μl Quick extraction buffer (Epicenter) and incubated in thermocycler (65 °C 15 min and 98 °C 5 min). One microliter of extracted genomic DNA was used to amplify the specific CFTR amplicon by Phusion Flash PCR Master Mix (Thermo Fisher). The PCR condition is as follows: 98 °C for 10 s, then 35 cycles of [98 °C for 1 s, 55 °C for 5 s, and 72 °C for 10 s], followed by a final 72 °C extension for 3 min. PCR products were purified by electrophoresis in a 1.5% agarose gel. The purified PCR products were identified for the targeted A-to-G conversion by sanger sequencing using the PCR forward primer. Single-cell clones with around 50% A-to-G conversion rate at target site were expanded from 24-well plate 2, further characterized, and cryopreserved. The primers for amplifying target CFTR genomic site are listed in the Supplementary Table.

**Electrophysiology assays**. Corrected 16HBE14ge W1282X cells and parental 16HBE14o- cells were seeded at a density of $4.5 \times 10^5$ cells/cm² onto HTS Transwell 24-well filter inserts (Corning, 3378) pre-coated with human collagen type IV (Sigma-Aldrich, C5533). Cells were grown as submerged cultures in MEM (Gibco, 11095) containing 10% FBS (Hyclone, SH30071.03) and 1% Pen/Strep, and incubated at 37 °C and 5% CO₂. After a total of 7 days, 16HBE cells typically formed electrically tight epithelia with a transepithelial resistance (Rt) of 200–600 Ω cm² and CFTR-mediated Cl− equivalent current (Ieq) was determined as described below.

Prior to functional (Ieq) studies, MEM was replaced with fresh HEPES-buffered (pH 7.4) solutions (assay buffer). A driving force for chloride ions was established through application of a basolateral to apical chloride ion gradient (see buffer composition below). Cell plates were mounted onto an automated robotic assay platform and equilibrated at ~36 °C for 90 min. After equilibration, transepithelial voltage (Vt) and resistance (Rt) were monitored at ~5 min intervals using a 24-channel transepithelial current clamp amplifier (TECC-24, EP Design, Bertem, Belgium). Electrode potential differences for each pair of Ag/AgCl voltage electrodes were also monitored at 5 min intervals by taking voltage measurements from a control plate with matching buffer solutions and 16HBE cells that were left untreated. Ieq was calculated from values of Vt and Rt using Ohm's law after correcting for series resistance and (electrode) voltage offsets unrelated to Vt. Ieq traces are plotted as mean ± SD ($n = 3$). The first four data points reflect baseline Ieq currents prior to sequential stimulation of CFTR with forskolin (10 μM) followed by VX-770/ivacaftor (1 μM). The last six data points were recorded in the presence of CFTR inhibitor CFTRinh-172 (20 μM). CFTR-mediated changes in Ieq, (the area under the curve (AUC) between forskolin and CFTRinh-172 addition) are used as a measure of functional CFTR surface expression or treatment-related functional rescue of mutant CFTR. Assay buffer: CFTR-mediated transepithelial currents were recorded using a Cl− concentration gradient. The basolateral solution contained (mM): 137 NaCl, 4 KCl, 1.8 CaCl₂, 1 MgCl₂, 10 HEPES and D-Glucose, adjusted to pH 7.4 with NaOH/HCl ([Cl−]total: 146.6 mM). The apical solution was matched to the basolateral except for (mM): 137 Na-gluconate replaced 137 NaCl ([Cl−]total: 9.6 mM).

**High throughput DNA sequencing of genomic DNA samples**. Genomic sites of interest were amplified from genomic DNA using the specific primers containing illumina forward and reverse adapters (listed in Supplementary Table). Twenty microliter PCR1 reactions were performed with 0.5 μM of each forward and reverse primer, 1 μl of genomic DNA extract or 300 ng purified genomic DNA, and 10 μl of Phusion Flash PCR Master Mix (Thermo Fisher). PCR reactions were carried out as follows: 98 °C for 10 s, then 20 cycles of [98 °C for 1 s, 55 °C for 5 s, and 72 °C for 10 s], followed by a final 72 °C extension for 3 min. After first round of PCR, unique Illumina barcoding reverse primer were added to each sample in a secondary PCR reaction (PCR 2). Specifically, 20 μl of a PCR reaction contained 0.5 μM of unique reverse Illumina barcoding primer pair and 0.5 μM common forward Illumina barcoding primer, 1 μl of unpurified PCR 1 reaction mixture, and 10 μl of Phusion Flash PCR Master Mix. The barcoding PCR2 reactions were carried out as follows: 98 °C for 10 s, then 20 cycles of [98 °C for 1 s, 60°C for 5 s, and 72 °C for 10 s], followed by a final 72 °C extension for 3 min. PCR 2 products were purified by 1% agarose gel using a QIAquick Gel Extraction Kit (Qiagen), eluting with 15 μl of Elution Buffer. DNA concentration was measured by Bioanalyzer and sequenced on an Illumina MiSeq instrument 150 bp, single-end) according to the manufacturer's protocols. Alignment of amplicon sequences to a reference sequence and calculation of the A-to-G conversion rate were performed according to reported script[1] (Supplementary Note). To analyze the frequency of bystander and corrected mutation at one CFTR allele, reads were aligned to the reference sequence to the defined editing window, and base calling of two tested bases were carried out simultaneously for each read. To calculate A5G9 or G5G9 frequency, we discarded the reads with low quality (Q < 30) for both of edited bases and used the equation: [frequency of specified point mutation] ÷ [total high-quality reads].

**Reporting summary**. Further information on research design is available in the Nature Research Reporting Summary linked to this article.

## Data availability

The source data underlying Figs. 1b-e, 2c–e, 3b, d, e and Supplementary Figs. 1a, c, e, f, 2a, b, 3a, b, 4c, d, e, 5a, c are provided as a Source Data File. The raw sequencing data have been submitted to the NCBI BioProject database (PRJNA616114 (https://www.ncbi.nlm.nih.gov/bioproject/616114) and PRJNA616118 (https://www.ncbi.nlm.nih.gov/bioproject/616118)). The all other data are available from the corresponding author upon reasonable request.

## Code availability

The script used to analyze editing efficiency is reported by Gaudelli and coworkers, and is provided in Supplementary Note.

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

## Acknowledgements

We thank C. Mello, P. Zamore, S. Wolfe, T. Flotte, and E. Sontheimer for discussions and E. Haberlin for editing the manuscript. We thank Dr. Markus Grompe (Oregon Health & Science University) for providing the Fah mice. We thank Y. Liu and E. Kittler in the UMass Morphology and Deep Sequencing Cores for support. W.X. was supported by grants from the National Institutes of Health (DP2HL137167, P01HL131471, and UG3HL147367), American Cancer Society (129056-RSG-16-093), the Lung Cancer Research Foundation, and the Cystic Fibrosis Foundation. This work was supported by DARPA HR0011-17-2-0049; U.S. NIH RM1 HG009490, R01 EB022376, U01 AI142756, and R35 GM118062; and HHMI (to D.R.L.). This work was supported in part by the Marble Center for Cancer Nanomedicine and a Cancer Center Support (core) Grant P30-CA14051 from the National Cancer Institute (to D.G.A.). G.A.N. was supported by the Helen Hay Whitney Fellowship. Q.W. was supported by a student fellowship by the China Scholarship Council.

## Author contributions

T.J. and W.X. designed the study. J.M.H. designed the mRNAs. T.J., J.M.H., K.C., Y.C., X.Z., Q.W., L.R., and Y.C. performed experiments and analyzed data. H.B., H.V., G.N., A.P.M., M.M., Z.W., D.G.A., and D.R.L. provided suggestions to the project. T.J. and W.X. wrote the manuscript with comments from all authors.

## Competing interests

D.R.L. is a consultant and co-founder of Editas Medicine, Pairwise Plants, Prime Medicine, and Beam Therapeutics, companies that use genome editing. W.X. is a consultant for the Cystic Fibrosis Foundation Therapeutics Lab. The other authors declare no competing interests.
