## [Peer Review File · Nature Communications]

Reviewers' Comments:

Reviewer #1:

Remarks to the Author:

The aim of this manuscript by Jiang et al. is to improve the base editing efficiency of ABE by chemical modifications of its mRNA and sgRNA. They found that 5-methoxyuridine modified ABE RA6.3 mRNA and modified sgRNA could mediate the A-to-G conversion rates comparable to RA6.3 and sgRNA expressed from plasmids. Furthermore, the authors proved its base editing function in pathogenic sites in vitro and in vivo. This manuscript performed comprehensive study and presented well. Nevertheless, I would like to see the following points addressed in the manuscript before its publication in Nature Communications.

1. In Fig.1c-e, the label should be consistent, and there should be an unmodified mRNA group as a control.
2. Will the bystander mutation of Fah mutation site cause codon changes? Please explain.
3. An unmodified mRNA group is necessary for analysis of cystic fibrosis cell model and the Tyrosinemia I mouse model. Otherwise, the authors cannot conclude the modified ABE is more efficient.
4. In the last paragraph of Results section, the authors claimed that "Importantly, at the bystander site (A6, Fig.3a), we observed a lower editing (0.096 ± 0.032) compared to our previous report ($1.9\pm 0.9\%$; Fig.3e), suggesting that the short half-life of ABE mRNA might minimize bystander conversions in vivo". Please give a professional analyses to see whether there is a significant difference between the two groups. Also, there missed a "%".

Reviewer #2:

Remarks to the Author:

In this manuscript, Xue and colleagues demonstrate that chemical modifications of the mRNA encoding a base editor and of the guideRNA can substantially improve the efficiency of base editing in hard to transfect somatic cells and in a mouse model of FAH.

The manuscript is clearly written, the experiments are well designed and described, and their results are correctly interpreted. The relevant scientific literature is cited, and the statistical analysis performed is, as far as I can judge, appropriate.

Base editing represents a promising approach to correct genetic defects in vivo and the manuscript will be of substantial interest to the broad readership of Nature Communications.

One aspect that is not discussed in the manuscript, but that would be useful to include in a revision, concerns off target effects of the system. Does the increased on-target activity promoted by the chemical modification also increases non-specific editing events?

General Response to Reviewers

We thank the reviewers for their constructive feedback, which has allowed us to greatly improve our manuscript. To address reviewers' concerns, we performed additional experiments that provide further support of robust base editing with our optimized, chemically modified RNA-encoded ABE system.

The following information has been added to address the major comments from reviewers:

1. Unmodified RNA data in **Fig. 2c**. 5moU mRNA achieved a significantly higher CFTR W1282X mutation correction rate than unmodified mRNA in 16HBEge cells.
2. Data shown in (**Supplementary Fig. 2**). We find that, at a relatively lower dosage, 5moU mRNA exhibits higher editing efficiency than unmodified mRNA at three tested genomic DNA sites in HEK293T cells.
3. Comparison of off-target effects raised by RNA and DNA-encoded ABE systems (**Supplementary Fig. 3**). Our data suggest that chemically-modified ABE mRNA did not exhibit significantly higher off-target editing than DNA-encoded ABE.
4. Addition of base editing references that were published since the first submission of our manuscript.

We also addressed all minor comments by the reviewers by performing additional analysis and adding relevant discussions.

Below is a point-by-point response to the specific comments of each reviewer.

Reviewers' comments:

Reviewer #1 (Remarks to the Author):

The aim of this manuscript by Jiang et al. is to improve the base editing efficiency of ABE by chemical modifications of its mRNA and sgRNA. They found that 5-methoxyuridine modified ABE RA6.3 mRNA and modified sgRNA could mediate the A-to-G conversion rates comparable to RA6.3 and sgRNA expressed from plasmids. Furthermore, the authors proved its base editing function in pathogenic sites in vitro and in vivo. This manuscript performed comprehensive study and presented well. Nevertheless, I would like to see the following points addressed in the manuscript before its publication in Nature Communications.

We thank the reviewer for the positive feedback.

1. In Fig.1c-e, the label should be consistent, and there should be an unmodified mRNA group as a control. We have modified our labels to be more consistent as ctrl, p_RA6.3+p_sgRNA, and 5moU-6.3+moderately modified_sgRNA.

According to previously reported study of Cas9 mRNA¹, we initially electroporated a high concentration (3 µg) of ABE mRNA in HEK293T cells to determine editing efficiencies in Fig.1c-e. To rigorously compare the editing efficiencies mediated by unmodified mRNA versus modified mRNA, we titrated the concentration (ranging from 0.0015µg to 0.5µg) of ABE mRNAs to measure their editing efficiencies. We then compared unmodified and modified mRNA at a relatively lower concentration (0.015µg) at the three genomic sites as tested in Fig.1c-e. These data are presented in **Supplementary Fig. 2**. We have added the following text to **Page 4** to summarize our results:

“Next, we compared the editing efficiency between unmodified and 5-methoxyuridine-modified ABE mRNA, when delivered with moderately modified sgRNA. We first measured A-to-G conversion rate at the same genomic site as in Figure 1c at different concentrations (ranging from 0.0015 to 0.5 µg) of unmodified-6.3 or 5moU-6.3 (Supplementary Fig.2a). We found that, at lower dosage (less than 0.015 µg), 5moU-6.3 shows ~1.5-fold higher editing efficiency compared to unmodified mRNA. Similarly, at two other genomic sites (same sites as in Figure 1d and e), 5moU-6.3 mediated substantially higher A-to-G conversion rates than unmodified mRNA (Supplementary Fig.2b). These data demonstrate a robust chemically modified RNA-encoded system for base editors.”

2. Will the bystander mutation of Fah mutation site cause codon changes? Please explain.

We thank the reviewer for raising this important question. We clarified the answer in the revised text, and cited the relevant paper, on **Page 6**:

“Notably, there is a bystander site within the editing window (A6, Fig.3a). The A-to-G conversion at this site changes a serine codon into alanine (S235A) in the FAH enzyme. Because S235 is near the FAH enzyme active site, A-to-G conversion at A6 site will not rescue the splicing defect but may affect enzyme activity.”

3. An unmodified mRNA group is necessary for analysis of cystic fibrosis cell model and the Tyrosinemia I mouse model. Otherwise, the authors cannot conclude the modified ABE is more efficient.

We thank the reviewer for this important comment. We added unmodified mRNA data for our experiments measuring correction rate of CFTR mutation in cells (**Fig. 2c**), and included the following text on **Page5**:

“We electroporated modified sgRNA with either unmodified-6.3 or 5moU-6.3 into 16HBEge cells to compare editing efficiency of unmodified versus 5-methoxyuridine-modified ABE mRNA. 5moU-6.3 achieved a significantly higher A-to-G conversion rate ($26.4\pm 7.40\%$) at A9 target site compared to unmodified-6.3 ($13.1\pm 0.509\%$).”

We did not analyze the efficiency of lipid nanoparticle-encapsulated unmodified ABE mRNA in correcting the *Fah* mutation in Tyrosinemia I mice model in this revision because 1) it is widely reported that unmodified mRNA can induce significant innate immune response in vivo^{2,4}, and 2) our data show that unmodified mRNA does not express well in cultured cells after lipid-based transfection (**Supplementary Fig. 1a, Fig. 1b**).

4. In the last paragraph of Results section, the authors claimed that “Importantly, at the bystander site (A6, Fig.3a), we observed a lower editing (0.096 ± 0.032) compared to our previous report ($1.9\pm 0.9\%$; Fig.3e), suggesting that the short half-life of ABE mRNA might minimize bystander conversions in vivo”. Please give a professional analyses to see whether there is a significant difference between the two groups. Also, there missed a “%”.

Thanks for this insightful comment. The missing “%” has been corrected. The statistical comparison in bystander site editing between the two groups is now presented in **Supplementary Fig. 5c**. We observed a statistically significant decrease in editing rate at the bystander site (A6) by RNA-encoded ABE in vivo. We have added the following text on **Page 6**:

*“Surprisingly, at this bystander site, we observed a significantly lower editing rate ($0.096\pm 0.032\%$) in LNP-treated mice compared to our previously reported rate ($1.9\pm 0.9\%$) ($P=0.0006$, **Supplementary Fig. 5c**)”*

Reviewer #2 (Remarks to the Author):

In this manuscript, Xue and colleagues demonstrate that chemical modifications of the mRNA encoding a base editor and of the guideRNA can substantially improve the efficiency of base editing in hard to transfect somatic cells and in a mouse model of FAH.

The manuscript is clearly written, the experiments are well designed and described, and their results are correctly interpreted. The relevant scientific literature is cited, and the statistical analysis performed is, as far as I can judge, appropriate.

Base editing represents a promising approach to correct genetic defects in vivo and the manuscript will be of substantial interest to the broad readership of Nature Communications.

We thank the reviewer for the positive feedback.

One aspect that is not discussed in the manuscript, but that would be useful to include in a revision, concerns off target effects of the system. Does the increased on-target activity promoted by the chemical modification also increases non-specific editing events?

We appreciate the reviewer’s insightful comments. We performed deep sequencing at top known off-target sites for two guide RNAs used in Fig1d, e. We added these data in **Supplementary Fig. 3** and included the following text to summarize our findings on **Page 4**:

“To compare the off-target effects induced by DNA versus modified RNA-encoded ABE systems, we analyzed A-to-G conversion rates at the top known off-target loci for the two guide RNAs used in Figure 1d and e. Deep sequencing data show that, overall, base editing rates at the off-target sites are low (<0.2% in all samples) (Supplementary Fig. 3a, b). A significant increase in A-to-G conversion rate was only detected at 2 of the 14 “A” sites in 5moU-6.3-treated cells compared to control or DNA-encoded ABE-treated cells. Our results suggest that chemically-modified ABE mRNA can improve on-target editing compared to DNA-encoded ABE without substantially increasing off-target effects.”

We also added text in the Discussion describing the limitation of our focused off-target analysis. On **Page7**:

“We did not observe significant off-target effects raised by our modified mRNA-encoded ABE at top known off-target sites for guide RNAs. However, some off-target events of base editor, on both the DNA and RNA level, may be independent of guide RNA sequence. Therefore, future work should perform unbiased screens to detect off-target editing by modified ABE mRNA at the whole genome and transcriptome levels.”

1. Vaidyanathan, S. et al. Uridine Depletion and Chemical Modification Increase Cas9 mRNA Activity and Reduce Immunogenicity without HPLC Purification. *Mol Ther Nucleic Acids* **12**, 530-542 (2018).
2. Kormann, M.S. et al. Expression of therapeutic proteins after delivery of chemically modified mRNA in mice. *Nat Biotechnol* **29**, 154-157 (2011).
3. Lutz, J. et al. Unmodified mRNA in LNPs constitutes a competitive technology for prophylactic vaccines. *NPJ Vaccines* **2**, 29 (2017).
4. Kauffman, K.J. et al. Efficacy and immunogenicity of unmodified and pseudouridine-modified mRNA delivered systemically with lipid nanoparticles in vivo. *Biomaterials* **109**, 78-87 (2016).

Reviewers' Comments:

Reviewer #1:

Remarks to the Author:

In this revised manuscript, the authors well addressed my concerns. I have no more question.

Reviewer #2:

Remarks to the Author:

The authors have addressed all my concerns. This is a highly meritorious work that deserves publication in Nature Communications.